# Adherence to COVID-19 preventive measures and associated factors in Ethiopia: A systematic review and meta-analysis

Gdiom Gebreheat[1,2]*, Ruth Paterson[2], Henok Mulugeta[3], Hirut Teame[4]

1 Department of Nursing, Adigrat University, Adigrat, Ethiopia, 2 School of Health and Social Care, Edinburgh Napier University, Edinburgh, United Kingdom, 3 Department of Nursing, Debre Markos University, Debre Markos, Ethiopia, 4 Department of Public Health, Adigrat University, Adigrat, Ethiopia

* gdiom7@gmail.com, gdiom.abady@napier.ac.uk

## Abstract

### Background

Reluctance to the COVID-19 preventive measures have been repeatedly reported in Ethiopia although compliance with these actions is the key step to minimize the pandemic's burden. Hence, this systematic review and meta-analysis aims to address the gap in the literature by determining the pooled magnitude of adherence to COVID-19 preventive measures and identifying its associated factors in Ethiopia.

### Materials and methods

The electronic databases used to search articles were PubMed/MEDLINE, CINAHL, Web of Science, ScienceDirect, Research4Life and other sources of grey literature including Google Scholar and World Health Organization (WHO) database portals for low- and middle-income countries. Full English-language articles published between 2019 and 2022 were eligible for the review and meta-analysis. Relevant data extracted and descriptive summaries of the studies presented in tabular form. The methodological quality of articles assessed using the Joanna Briggs Institute (JBI) quality assessment tool. The pooled magnitude of adherence determined by applying a random-effects model at a 95% CI.

### Results

Of 1029 records identified, 15 articles were included in the systematic review and 11 were selected for meta-analysis. The pooled estimate of adherence to COVID-19 preventive measures in Ethiopia was 41.15% (95% CI:32.16–50.14%). Furthermore, perceived COVID-19 disease severity (AOR:1.77, 95% CI: (1.40–2.25)), attitude (AOR:1.85, 95% CI: (1.36–2.53)) and knowledge (AOR:2.51, 95% CI: (1.67–3.78)) to COVID-19 preventive measures showed significant association with adherence to COVID-19 preventive measures.

**Data Availability Statement:** The data used to support the findings of this study are included in the manuscript as supportive information.

**Funding:** The authors received no specific funding for this work.

**Competing interests:** The authors have declared that no competing interests exist.

## Conclusion

The magnitude of adherence to COVID-19 preventive measures in Ethiopia appeared to be low. Therefore, the government of Ethiopia and other stakeholders should mobilize resources to improve the adherence level of the community to the COVID-19 preventive measures and decrease public fatigue.

## Introduction

The current pandemic, COVID-19 disease, is a highly contagious viral infection caused by novel coronavirus called severe acute respiratory syndrome coronavirus 2 (SARS-CoV-2). Person-to-person spread is the main means of transmission, when any infected person sneeze or cough respiratory droplets of the virus, and these droplets enter the lungs of a nearby person via inhalation. Besides, environmental contamination is another way to spread the virus, resulting in an unprecedented threat to global health and well-being. Patients infected from SARS-CoV-2 infection often presented with dry cough, fever, sputum production and shortness of breath and upper airway congestion [1,2].

The WHO has recommended multiple COVID-19 prevention and control measures including proper hand washing, physical distancing, covering mouth and nose when coughing and sneezing, avoiding touching face and staying at home [3]. Several countries have implemented the WHO recommendations to prevent and control COVID-19 infection [4]. The earlier announcement of lockdown and the stricter the adherence was believed to lead to fewer infected total cases and deaths. Hence, it was expected to accelerate the containment of the virus and lessen the consequences of the mitigating measures [5]. Unfortunately, globally, as of 1 August, 2022, there were more than 577 million confirmed cases of COVID-19, including 6.4 million deaths, reported to WHO [6].

In Ethiopia, following the confirmation of the first case of COVID-19 in March 2020, the Ethiopian ministry of health and public health institute have taken several initiatives to decrease the burden of COVID-19 [7]. Among these, hand hygiene, facemask and social distancing were the primary three preventive measures that the government communicated to the community through various media platforms [8]. Furthermore, measures were imposed to close schools, restrict major gatherings and movements of people, and even lockdown. Despite this, the preventive measures were being ignored by the community and leaders at different level [9].

Poor adherence towards COVID-19 mitigation measures has continued as an escalating problem in Ethiopia [10]. According to the study conducted among Hossana residents, nearly half of them had poor adherence to the COVID-19 preventive measures [11]. Likewise, in a recent study in the capital of Ethiopia, Addis Ababa, nearly 40% of the community has shown poor implementation of COVID-19 preventive measures [12]. Surprisingly, in a study of Oromia region of Ethiopia, the overall adherence level of the community to the recommended safety measures of COVID-19 was 8.3% [13], which seems far lower than the other studies. The main barriers to effective implementation of public health measures were resistance to change, lack of community engagement, negligence, insufficient training for front line workers, poor law enforcement, poor supportive supervision, and lack of continuous community awareness creation [14].

However, there is no pooled evidence on the magnitude of adherence to COVID-19 preventive measures and its associated factors in Ethiopia. Thus, this meta-analysis aims to estimate the magnitude of adherence to COVID-19 preventive measures and its associated

factors. More importantly, the systematic review and meta-analysis results will help decision-makers to plan and implement effective action against the COVID-19 pandemic.

## Materials and methods

### Search strategies

Research articles were accessed through electronic web-based database searches and reference list reviews using the Preferred Reporting Items of Systematic Reviews and Meta-Analysis (PRISMA) checklist guidelines [15]. Literature that reported adherence status to COVID-19 preventive measures and/or its associated factors in Ethiopia were searched from both main electronic databases and grey literature sources. The electronic databases used to search articles were PubMed/MEDLINE, ScienceDirect, Web of Science, CINAHL, Research4Life and other World Health Organization (WHO) database portals for low- and middle-income countries. In addition, the researchers found related articles through a desk review of the doctoral dissertations available at Ethiopian university libraries and institutional repositories, and from reviewing the reference lists of related articles. Electronic database searches were conducted from July 20, 2022, to July 23, 2022. The main terms used during electronic database search were: ("Adherence" OR "Compliance" OR "Associated factors" OR "Determinants" OR "Predictors") AND ("COVID-19 prevention measures" OR "COVID-19 preventive measures") AND (Ethiopia). Please see S1 Table for a detail article searching process, terms used in each database and search results.

### Inclusion and exclusion criteria

All English-language, full-text, original research articles conducted in Ethiopia from January1, 2019 to July 23, 2022 and published in peer-reviewed journals or filed as completed dissertations were considered for inclusion. Moreover, the article should measure the adherence level and/or associated factors of COVID-19 preventive measures in Ethiopia. However, case series, opinion papers and reports were excluded.

### Studies screening and selection process

All electronic search results were transferred into Mendeley reference manager software version 1. 19.8. Next, we organized all these articles into a single folder for duplicate citation removal and further management of articles. After removing duplicate citations with the software, two authors (GG, RP) independently screened the articles, based on preset eligibility criteria. The article screening process had three sequential stages, title, abstract and full-text screening. Through title screening, studies entitled with terms directly/indirectly measure the adherence level and/or associated factors of COVID-19 preventive measures in Ethiopia, were selected for abstract screening. And, in abstract screening, articles were read their abstract if they could measure either of the review and meta-analysis outcomes. Consequently, full-text screenings were carried out with four independent authors. The final decision whether to include an article were reached on the consensus of all the authors. The screening and selection of articles were guided according to the PRISMA guideline (Fig 1).

### Critical appraisal of studies

Quality of studies was critically evaluated for the validity of results. The methodological quality of the papers was assessed using the JBI quality assessment checklists for cross-sectional analytical studies [16]. This JBI critical appraisal checklist has eight elements, which mainly addresses the methodological area of each article. It focused on the appropriateness of the statistical

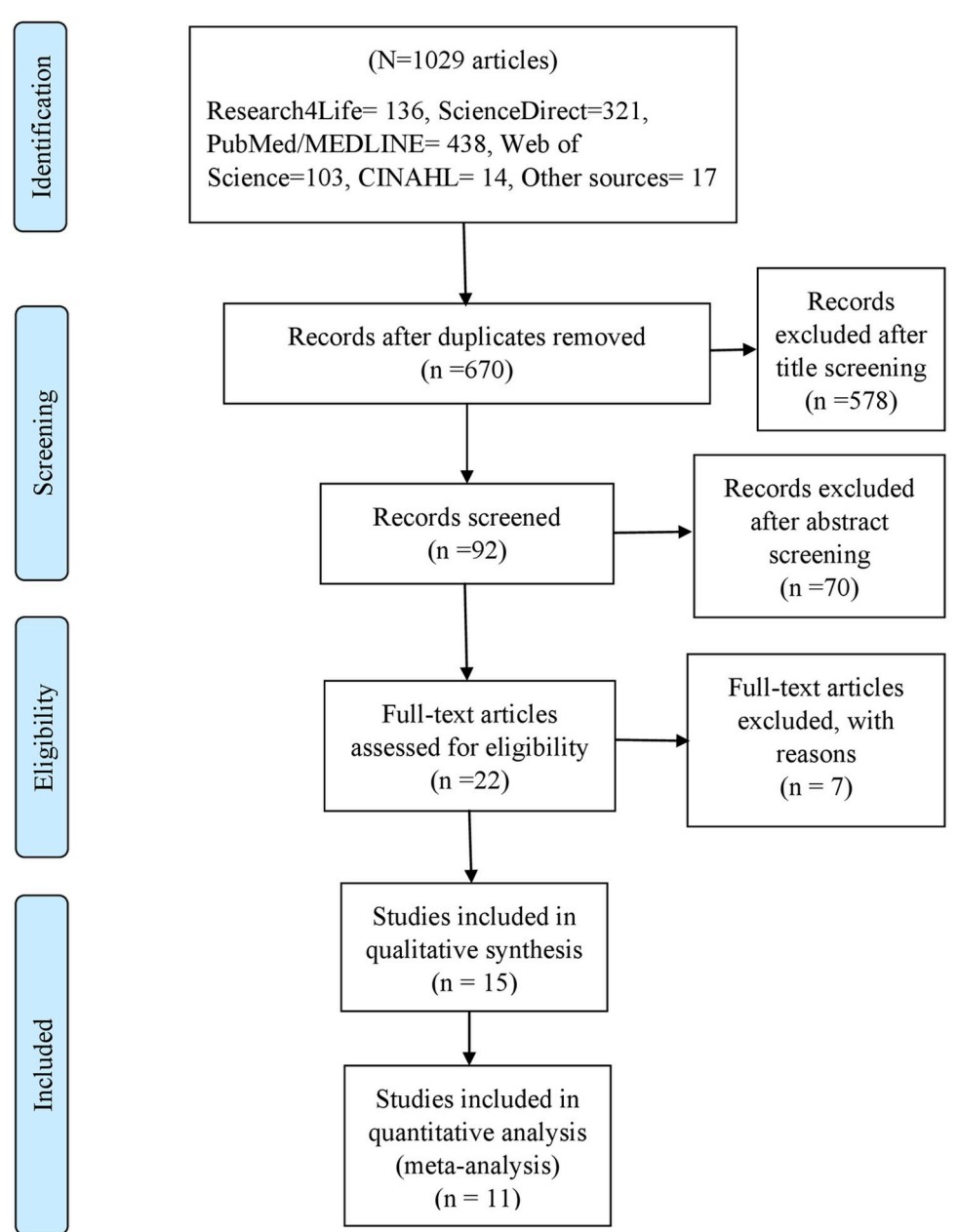

**Fig 1. A PRISMA flow chart that shows the process of article selection.**

analysis, objective, inclusion criteria, study population and setting, exposure and outcomes measurement, and management of confounding factors. The evaluation and decision of each article was finalized on the consensus of all authors. Accordingly, articles with positive answers (yes) for more than 50% of the eight-elemental checklist (i.e., yes for five or more) were included in this systematic review and meta-analysis (S2 Table).

## Data extraction

After all authors agreed on the articles to be included in the review and meta-analysis, we set an extraction template in the Microsoft Excel sheet. The study description in Table 1 was

**Table 1. Characteristics and main findings of the articles included in the systematic review and meta-analysis.**

| Author, Year | Aim | Study population | Study design | Sample size | Response rate (%) | Adherence level (%) | Factors associated with adherence to COVID-19 preventive measures |
|---|---|---|---|---|---|---|---|
| Abeya et al., 2021 [13] | To assess the level of adherence to COVID-19 preventive measures and associated factors in the study area | Community | Cross-sectional | 2751 | 95.5 | 8.3 | Age, illiteracy, read and write, attended primary, occupation and knowledge were factors associated with level of adherence to COVID-19 preventive measures |
| Asnakew et al., 2020 [30] | To assess the community's level of risk perception of COVID-19, their compliance with recommended precautionary measures, and factors that influence compliance behavior | Community | Cross-sectional | 521 | NA | NA | Being female, higher perceived effectiveness of recommended preventive measures, and higher perceived reliability of media facilitated compliance with preventive measures. Increasing age, being single, lower education level and living at a lower administrative level were barriers to be compliant. |
| Azene et al., 2020 [10] | To assess the community's adherence towards COVID-19 mitigation strategies and its associated factors | Community | Cross-sectional | 635 | 98.1 | 51.04 | Being female, good level of information exposure, good knowledge about COVID-19, favourable attitude towards COVID-19 prevention measures and high-risk perception of COVID-19 |
| Bante et al., 2021 [25] | To assess communities' adherence with COVID-19 preventive measures and its associated factors | Community | Cross-sectional | 648 | 99.4 | 12.3 | Urban residence, favourable attitude towards COVID-19 prevention measures and fear of stigma due to COVID-19 |
| Etafa et al., 2021 [22] | To assess healthcare workers' compliance with measures to prevent COVID-19, and its potential determinants in public hospitals | Health professional | Cross-sectional | 422 | 95.3 | 22 | Spending most of caring time at bedside, receiving training on infection prevention/COVID-19, reading materials on COVID-19 and getting support from hospital management |
| Hailu et al., 2021 [29] | To assess the compliance, barriers, and facilitators to social distancing measures for the prevention of COVID-19 in Northwest Ethiopia | Community | Cross-sectional | 425 | 94.4 | NA | Poor compliance with social distancing measures. Age, older persons more likely than younger to comply with social distancing guidelines. |
| Kayrite et al., 2020 [21] | To measure the compliance with COVID-19 preventive and control measures | Community | Cross-sectional | 324 | 97 | 55.50 | NA |
| Kebede et al., 2021 [27] | To assess healthcare provider's adherence to COVID-19 preventive practices during childbirth in northwest Ethiopia | Health professional | Cross-sectional | 406 | 96.4 | 46.1 | Healthcare providers who had job satisfaction, had smartphone and/or computer, ever received training on infection prevention, earned higher monthly income, and worked at health facility in the urban area had a significant association with adherence to COVID-19 preventive practices. |
| Keleb et al., 2021 [19] | To determine the magnitude of compliance and associated factors of personal protective equipment utilization and hand hygiene practice among healthcare workers in public hospitals of South Wollo Zone, Northeastern Ethiopia. | Health professional | Cross-sectional | 489 | 96.8 | NA | About 32 and 22.3% of healthcare workers were compliant with personal protective equipment utilization and hand hygiene practice, respectively. Feedback for safety, training on COVID-19 prevention, and perception to infection risk were significant factors of good compliance with personal protective equipment utilization. |

(*Continued*)

**Table 1.** (Continued)

| Author, Year | Aim | Study population | Study design | Sample size | Response rate (%) | Adherence level (%) | Factors associated with adherence to COVID-19 preventive measures |
|---|---|---|---|---|---|---|---|
| Shewasinad et al., 2021 [24] | To identify the predictors of adherence to COVID-19 prevention measure among | Community | Cross-sectional | 683 | 100 | 44.10 | Perceived usefulness of safety measures, absence of perceived barriers to COVID-19 safety measures and perceived non susceptibility of COVID-19 |
| Silesh et al., 2021 [26] | To assess compliance with COVID-19 preventive measures among pregnant women attending antenatal care at public facilities of Debre Berhan town, Ethiopia | Community (pregnant mothers) | Cross-sectional | 402 | 98.5 | 56.1 | Maternal age, husband educational status, chronic disease, and knowledge were significant predictors to have good compliance with COVID-19 preventive measures. |
| Temesgan et al., 2022 [28] | To assess adherence to COVID-19 preventive practice and associated factors among pregnant women in Gondar city, northwest Ethiopia | Community (pregnant mothers) | Cross-sectional | 678 | 97.8 | 44.8 | Age, education, having ANC follow up and adequate knowledge towards COVID-19 were significantly associated with good adherence to COVID-19 preventive practice |
| Temesgen et al., 2021 [11] | To determine adherence to covid-19 prevention measures | Community | Cross-sectional | 384 | 98.2 | 50.4 | Age < 20 years, married, household size 7 and above and having information about the complication of COVID-19 |
| Zenbaba et al., 2021 [23] | To assess the compliance towards COVID-19 preventive measures and associated factors | Health professional | Cross-sectional | 660 | 99 | 49.9 | Working in referral hospital, age 24 or younger years old, 3–6 years of work experience, good knowledge regarding COVID-19 preventive measures, knowing the presence COVID-19 Prevention Committee, having functional handwashing facilities and continuous water supply at workplace |
| Zewude et al., 2021 [20] | To examine compliance to personal protective behavioral recommendations to contain the spread of COVID-19 among urban residents engaged in the informal economic activities in Wolaita Sodo town, Southern Ethiopia | Community | Cross-sectional | 384 | 100 | NA | Regular wearing of a mask was significantly associated with regular attendance of the media regarding the preventive mechanisms of COVID-19, knowledge of someone ever infected by COVID-19, the belief that COVID-19 causes a severe illness, and perception of the likelihood of dying as a result of infection by COVID-19 |

formulated to summarize the study design, study setting, sample size, aim, key finding (magnitude of adherence to COVID-19 preventive measures), and secondary outcome (associated factors with Adherence to COVID-19 preventive measures). Data extraction was carried out by two authors (GD and RP) and double checked by the other authors. The extracted numerical data were documented and stored in a Microsoft Excel separate sheet (S3 Table).

## Outcome of interest

The primary outcome of interest was the pooled magnitude of adherence to COVID-19 preventive measures in Ethiopia. The magnitude of adherence was measured as the number of adhering study subjects divided by the total sample size multiplied by 100. Secondly, we have also pooled the odds ratio of each factor to see if there was a statistical association with adherence to COVID-19 preventive measures in Ethiopia.

## Data analysis

The raw data in the Microsoft Excel spreadsheet template was transferred to STATA™ version 16 software for analysis. A pooled magnitude of adherence to COVID-19 preventive measures in Ethiopia was estimated at a 95% confidence interval (CI). Further, we conducted a regional subgroup analysis. Also, a pooled odds ratio of different variables was calculated using a RevMan version 5.4.1 to check if there was an association between independent variables and the dependent variable (adherence to COVID-19 preventive measures). The heterogeneity of study outcomes was assessed using the $I^2$ statistic [17]. Accordingly, studies with high heterogeneity were estimated using a random-effects model, and fixed effect model was run in variables showed low heterogeneity (<50%). Parallelly, a publication bias was checked using a funnel plot asymmetry and Egger's and Begg-Mazumdar Rank correlation tests [18]. Eventually, the statistical analysis and the results were double-checked by all authors.

## Results

### Identification and description of studies

A total of 1029 citations were collected through electronic database search and other sources (Fig 1). Of these, we excluded 359 items due to duplication. From the remained 670 collections, 578 items were excluded through title screening, while 70 were excluded after the abstract screening. Next, 22 full articles were reviewed according to the predefined eligibility criteria. Eventually, 15 articles were found fully eligible for systematic review, of which 11 articles were included in meta-analysis. All of them were conducted using a cross-sectional study design [10,11,13,19–30]. Nearly half (n = 7) of these studies were conducted in Amhara region [10,19,24,27–29,31], four articles in SNNP (Southern Nations, Nationalities, and Peoples') region [11,20,21,25], three in Oromia region [13,22,23] and the remained one study was conducted in Addis Ababa [30]. The maximum sample size recorded was 2751 subjects [13], while the minimum was 324 [19]. Furthermore, majority of the studies had more than 95% response rate (Table 1).

### Quality appraisal of the review

The JBI quality assessment tools were used to evaluate the methodological quality of the articles, based on the consensus of the two evaluators (RP, GD). We included studies with clear eligibility criteria for inclusion in the sample, a detailed description of the context, a reliable and valid measure of exposure, and adequate statistical analysis. Both authors agreed that articles with $\geq$ 50% of the total score to be included in the systematic review and meta-analysis. As a result, 15 studies were of high methodological quality for the primary outcome of interest [10,11,13,19–30] (S2 Table).

### Publication bias

Publication bias was evidenced on both the funnel plots of precision asymmetry and the Egger's test of the intercept. We run a trim and fill analysis in the random-effects model [32]. The magnitude estimates did not differ significantly between the initial and, trim and fill models (Figs 2 and 3).

### Adherence to COVID-19 preventive and control measures

Fifteen articles [10,11,13,19–30] discussed the participants' level of adherence to COVID-19 preventive measures in Ethiopia. Of these, four articles [19,20,29,30] discussed the adherence level of participants to each preventive measures separately while the remained 11 articles

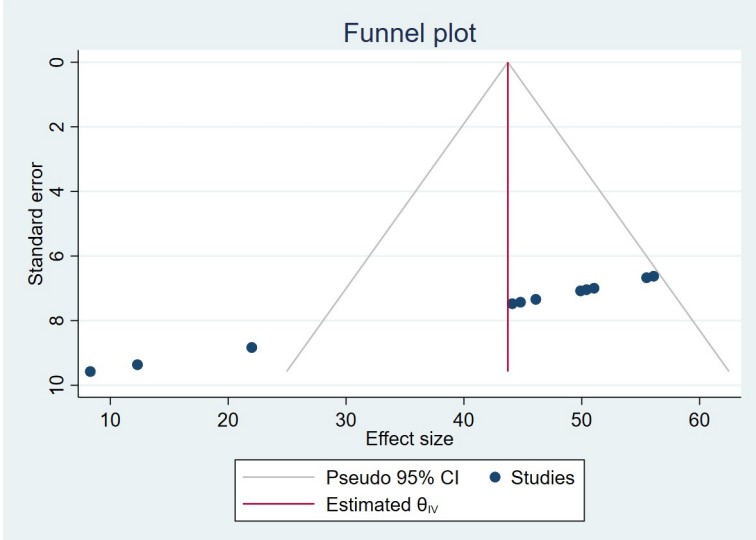

**Fig 2. Funnel plot with pseudo 95% confidence interval limits.**

[10,11,13,21–28] estimated the overall compliance or adherence level. In two articles, the compliance percentage to hand hygiene was 22.3 [19] and 97.1 [30]. W. Hailu, et al. reported that 55.4% of the participants had poor compliance with social distancing measures [29]. In another article, 35.4% of respondents reported to be adherence with regular mask wearing [20]. Furthermore, 11 articles [10,11,13,21–28] were included to estimate the pooled magnitude of adherence to COVID-19 preventive measures in Ethiopia. The magnitude of adherence

**Fig 3. Filled funnel plot with pseudo 95% confidence interval.**

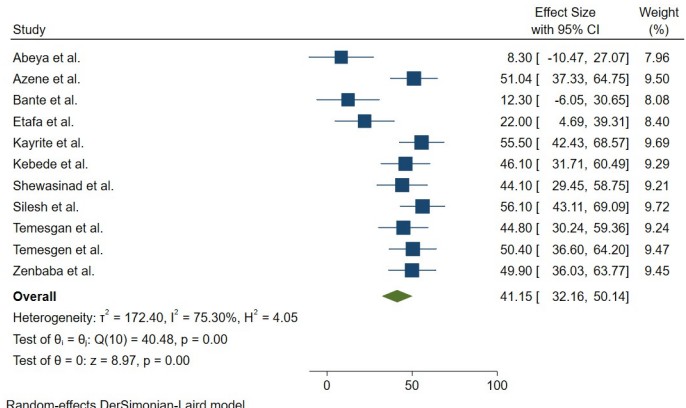

**Fig 4. Forest plot studies assessing the magnitude of adherence to COVID-19 preventive measures in Ethiopia.**

to COVID- 19 preventive measures in Ethiopia was ranged from 8.3% in Oromia [13] to 56.1% in Amhara [26]. Hence, the $I^2$ statistic for heterogeneity has shown significant differences between the studies ($I^2 = 99$%, p<0.05). Therefore, we have decided to fit DerSimonian and Laird random-effects model to estimate the pooled magnitude of adherence to COVID-19 preventive measures. Our decision was based on the theoretical assumptions that the heterogeneity (difference between the studies) might be because of the settings and socio-economic contexts [33,34]. The model also shows the weight of each study as per its sample size and effect size [35]. Accordingly, highest weight was reported in Silesh et al, 9.72% [26] while the lowest weight was recorded in a study conducted by Abeya et al., 7.96% [13] (Fig 4). The pooled estimate of adherence to COVID-19 preventive measures in Ethiopia was 41.15% (95% CI:32.16–50.14%) (Fig 4).

Also, the sub-group analysis of study setting showed that the pooled magnitude of adherence to COVID-19 measures in Amhara, SNNP and Oromia regions of Ethiopia was, 48.8% (95% CI:42.59–55.12%), 40.22% (95% CI:16.46–63.97%) and 27.35% (95% CI:2.24–52.46%), respectively (Fig 5).

## Factors associated with adherence to COVID-19 preventive measures in Ethiopia

Twelve articles have discussed about the associated factors of adherence to COVID-19 preventive measures in Ethiopia [10,11,13,19,20,23–26,28–30]. Of these, 8 articles were included in our meta-analysis to identify the associated factors of adherence to COVID-19 preventive measures in Ethiopia [10,11,13,23–26,28]. Sex [10,11,23–25,30], age [11,24,25,29], perceived COVID-19 severity [10,11,19,20,24,25], attitude [10,11,13,25–26] and knowledge [10,11,13,23,25,26,28] to COVID-19 preventive measures were found to have a statistically significant association with adherence to COVID-19 preventive measures (Figs 6–10).

**Sex.**   In this meta-analysis, the sex of the study participants was found to be a statistically significant factor in the adherence level of COVID-19 preventive measures. Male participants were 36% less likely to adhere to COVID-19 preventive measures than female participants (AOR:0.64, 95% CI: (0.54–0.78)). The heterogeneity between these studies was low (Fig 6).

**Age.**   Also, age was another factor that had a statistically significant correlation with the adherence status of study participants to COVID-19 preventive measures in Ethiopia. People who were younger than 40 years old had 1.6 odds of adherence to COVID-19 preventive measures (AOR:1.6, 95% CI: (1.04–2.46)) (Fig 7).

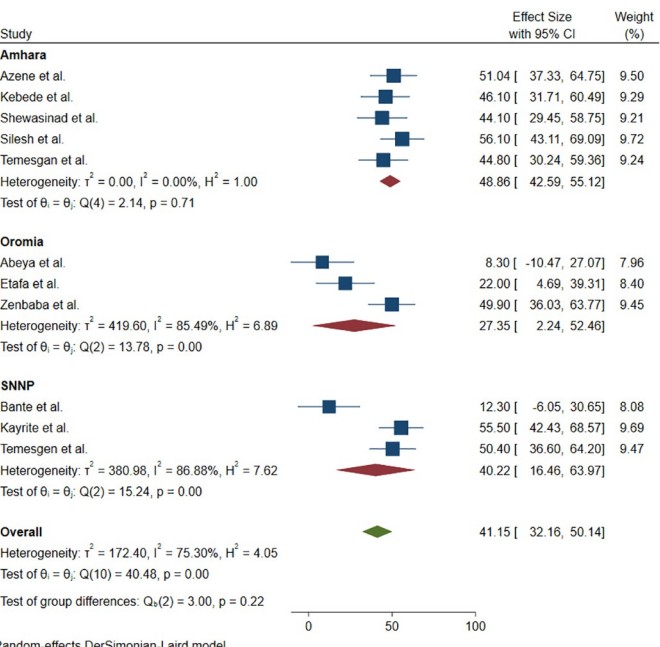

**Fig 5. Forest plot of studies assessing region-based magnitude of adherence to COVID-19 preventive measures in Ethiopia.**

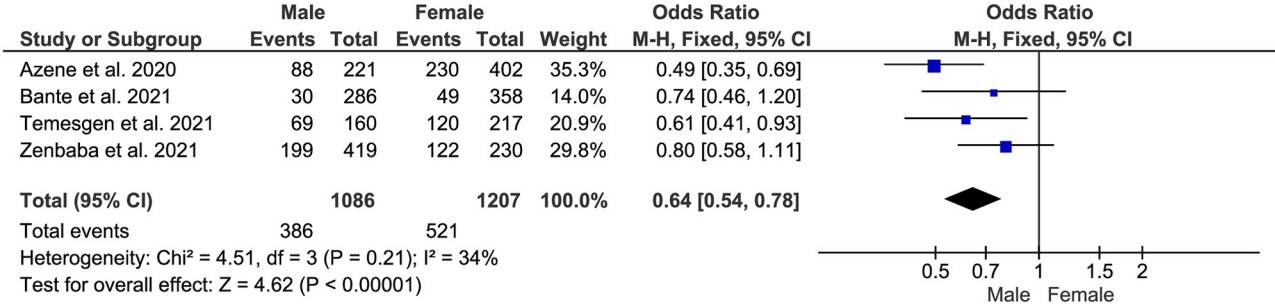

**Fig 6. Association between sex and adherence to COVID-19 preventive measures in Ethiopia.**

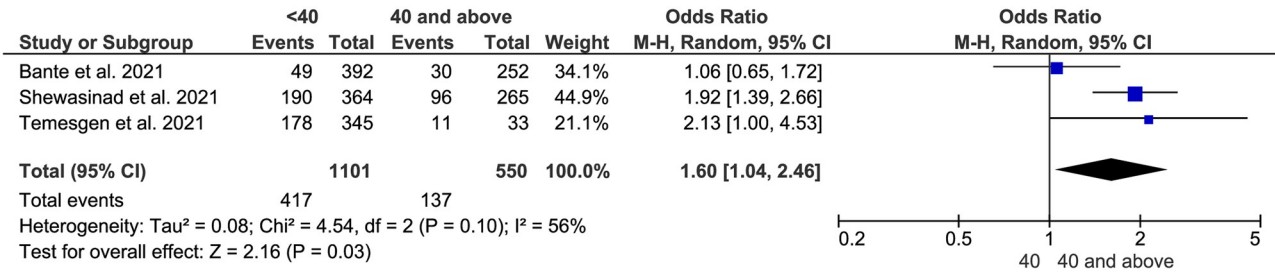

**Fig 7. Association between age and adherence to COVID-19 preventive measures in Ethiopia.**

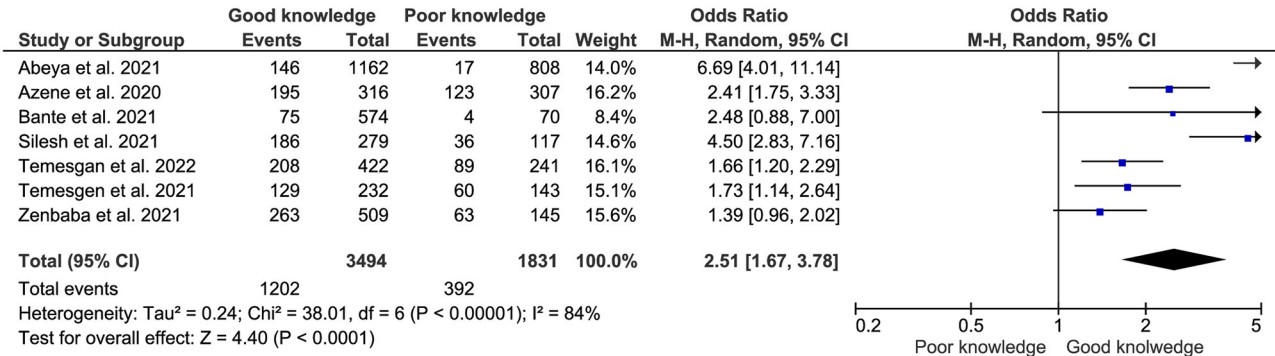

**Fig 8. Association between knowledge and adherence to COVID-19 preventive measures in Ethiopia.**

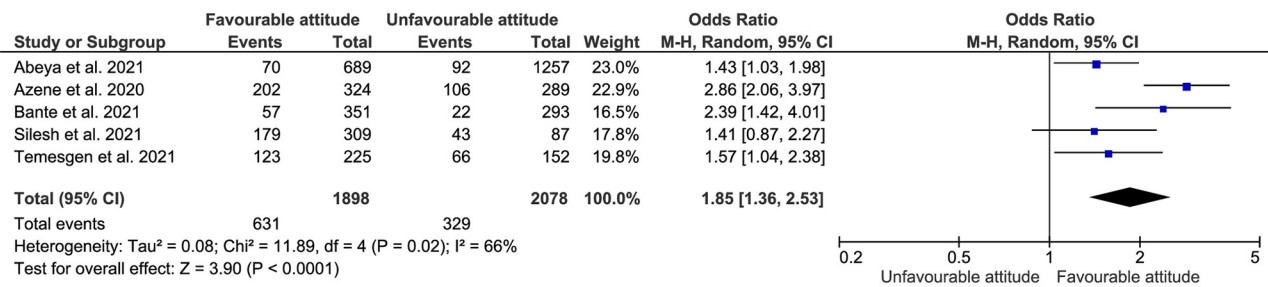

**Fig 9. Association between attitude and adherence to COVID-19 preventive measures in Ethiopia.**

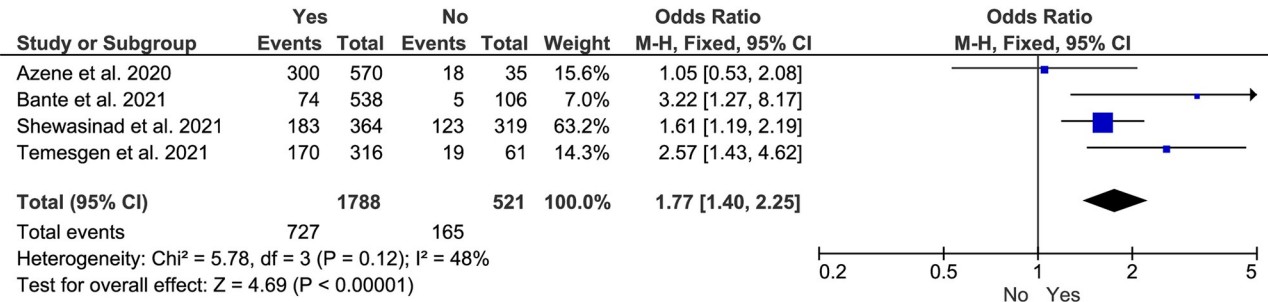

**Fig 10. Association between perceived disease severity and adherence to COVID-19 preventive measures in Ethiopia.**

**Knowledge of COVID-19 preventive measures.** The present meta-analysis also found that people's knowledge about COVID-19 preventive measures significantly associated with their adherence status to the COVID-19 preventive measures. Those who had good knowledge about COVID-19 were 2.51 more likely to adhere to COVID-19 preventive measures than those who had poor knowledge, (AOR:2.51, 95% CI: (1.67–3.78)) (Fig 8).

**Attitude towards COVID-19 preventive measures.** People who had a favourable attitude towards COVID-19 preventive measures were 1.85 times more likely to adhere to the preventive measures, (AOR:1.85, 95% CI: (1.36–2.53)) (Fig 9).

**Perceived severity of COVID-19 disease.** Perceived COVID-19 severity was also another significant variable that correlated with the peoples' adherence status to the COVID-19 preventive measures in Ethiopia. People who had perceived the severity of COVID-19 infection

were 1.77 times more likely to adhere to the COVID-19 preventive measures than those who had not this perception (AOR:1.77, 95% CI: (1.40–2.25)) (Fig 10).

## Discussion

This systemic review and meta-analysis attempted to estimate the pooled magnitude of adherence to COVID-19 preventive measures and its associated factors in Ethiopia. We found 41.15% of the people were adhered to the COVID-19 preventive measures in Ethiopia. This is consistent with a previous review in the country where the pooled level of health professional' practice, towards WHO COVID-19 management and personal protection recommendations, estimated to be 40.3% [36]. This low level of adherence to COVID-19 preventive measures could be related with public fatigue, ignorance, misinformation, personal or social norms and perceived behavior control [37–40].

Another aim of this study was to identify the associated factors of adherence to COVID-19 preventive measures. Accordingly, sex, age, perceived COVID-19 severity, attitude, and knowledge of COVID-19 preventive measures were found to have a statistically significant association with the adherence to COVID-19 preventive measures.

Regarding sex, male participants were 36% less likely to adhere to COVID-19 preventive measures than female participants (AOR:0.64, 95% CI: (0.54–0.78)). This finding is consistent with previous studies conducted in Canada, China, Brazil, Uganda and Somalia, where male participants had had lower compliance with the COVID-19 preventive measures [41–45]. In the context of Ethiopia, men are the one often who runs the outdoor day-to-day activities that may lead to break the COVID-19 preventive measures.

People who were younger than 40 years old had 1.6 odds of adherence to COVID-19 preventive measures (AOR:1.6, 95% CI: (1.04–2.46)). We believe that Ethiopia has a higher educated adult population than it had before decades. Therefore, related with the recent digital technology advancements, the young generation might have better awareness and adherence to COVID-19 preventive measures than the old population. Conversely, studies from Brazil and Switzerland indicated that young adults are non-compliant [41,46]. This controversy might be because of the socio-demographic differences of study participants between Ethiopia and, Brazil and Switzerland.

In our meta-analysis, those who had good knowledge about COVID-19 preventive measures were 2.51 more likely to adhere to COVID-19 preventive measures than those who had poor knowledge, (AOR:2.51, 95% CI: (1.67–3.78)). This finding is complementary to that of the Democratic Republic of the Congo [47]. In connection, knowledge of restrictions can also predict positive attitudes towards restrictions and increase perceived ability to adhere to the mitigating measures as well [48]. Therefore, in this review, people who had a favourable attitude towards COVID-19 preventive measures had had 1.85 times more likely to adhere to the preventive measures (AOR:1.85, 95% CI: (1.36–2.53)). In line with this, a study from Iran established the positive relationship between positive attitude towards the effectiveness of preventive measures and adherence to them [49]. Therefore, the participants' awareness and attitude could have an impact on their level of compliance to the rules and regulation of COVID-19 preventive measures.

Our finding also revealed that people who perceived the severity of COVID-19 had 1.77 odds of adherence to the COVID-19 preventive measures, (AOR:1.77, 95% CI: (1.40–2.25)). In previous studies, perceived susceptibility to COVID-19 infection and perceived severity of health-related consequences were linked to engagement in disease-preventive behaviors. Consequently, there is a possibility of a positive relationship between perceived severity and adherence [50–53]. Also, according to previous Health Belief Model (HBM)-based studies, people's

perception of the seriousness of having COVID-19 infection can dictate them to comply with recommended preventive measures [54–57].

## Limitations

Our systematic review and meta-analysis had some limitations. First, all included studies were cross-sectional by design. Instead, it would have been more impactful if studies with variety of design had been included in the review. Secondly, the presence of heterogeneity between studies may not be supportive to draw inclusive inference about the general population.

## Conclusion

This systematic review and meta-analysis found that the level of adherence to COVID-19 preventive measures in Ethiopia was low, below 50%. Furthermore, sex, age, perceived COVID-19 severity, attitude and knowledge to COVID-19 preventive measures were found to have a statistically significant association with adherence to COVID-19 preventive measures. Therefore, the government of Ethiopia and other stakeholders should mobilize resources to improve the adherence level of the community to the COVID-19 preventive measures and decrease public fatigue.

## Supporting information

**S1 Table. Search strategy.**
(DOCX)

**S2 Table. Critical appraisal.**
(DOCX)

**S3 Table. Extracted raw data.**
(XLSX)

**S1 Appendix. PRISMA checklist.**
(DOCX)

**S2 Appendix. STATA dataset for adherence level.**
(DTA)

**S3 Appendix. RevMan dataset for factors associated with adherence to COVID-19 preventive measures.**
(RM5)

## Acknowledgments

We thank Adigrat University and Edinburgh Napier University for allowing us to access their databases.

## Author Contributions

**Conceptualization:** Gdiom Gebreheat, Hirut Teame.

**Data curation:** Gdiom Gebreheat, Henok Mulugeta, Hirut Teame.

**Formal analysis:** Gdiom Gebreheat, Hirut Teame.

**Investigation:** Gdiom Gebreheat, Hirut Teame.

**Methodology:** Gdiom Gebreheat, Henok Mulugeta, Hirut Teame.

**Project administration:** Gdiom Gebreheat.

**Resources:** Gdiom Gebreheat.

**Software:** Gdiom Gebreheat, Hirut Teame.

**Validation:** Gdiom Gebreheat, Ruth Paterson, Henok Mulugeta, Hirut Teame.

**Visualization:** Gdiom Gebreheat, Ruth Paterson, Hirut Teame.

**Writing – original draft:** Gdiom Gebreheat, Ruth Paterson, Hirut Teame.

**Writing – review & editing:** Gdiom Gebreheat, Ruth Paterson, Henok Mulugeta, Hirut Teame.

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
