## [Decision Letter · Decision Letter 0]

20 Jul 2022

PONE-D-21-32366Adherence to COVID-19 prevention measures and associated factors in Ethiopia: A systematic review and meta-analysisPLOS ONE

Dear Dr. Gebreheat,

Thank you for submitting your manuscript to PLOS ONE. After careful consideration, we feel that it has merit but does not fully meet PLOS ONE’s publication criteria as it currently stands. Therefore, we invite you to submit a revised version of the manuscript that addresses the points raised during the review process.

I would like to sincerely apologize for the delay you have incurred with your submission. It has been exceptionally difficult to secure reviewers to evaluate your study. We have now received two completed reviews; the comments are available below. The reviewers have raised significant scientific concerns about the study that need to be addressed in a revision.

Please revise the manuscript to address all the reviewer's comments in a point-by-point response in order to ensure it is meeting the journal's publication criteria. Please note that the revised manuscript will need to undergo further review, we thus cannot at this point anticipate the outcome of the evaluation process.

We look forward to receiving your revised manuscript.

Kind regards,

Miquel Vall-llosera Camps

Senior Editor

PLOS ONE

Journal Requirements:

5. Please upload a new copy of Figures 3-9 as the detail is not clear. Please follow the link for more information: https://blogs.plos.org/plos/2019/06/looking-good-tips-for-creating-your-plos-figures-graphics/" https://blogs.plos.org/plos/2019/06/looking-good-tips-for-creating-your-plos-figures-graphics/

6. Please include a copy of Table 1 which you refer to in your text on page 7.

Reviewers' comments:

Reviewer's Responses to Questions

**Comments to the Author**

1. Is the manuscript technically sound, and do the data support the conclusions?

Reviewer #1: Partly

Reviewer #2: Yes

2. Has the statistical analysis been performed appropriately and rigorously? 

Reviewer #1: Yes

Reviewer #2: Yes

3. Have the authors made all data underlying the findings in their manuscript fully available?

Reviewer #1: Yes

Reviewer #2: Yes

4. Is the manuscript presented in an intelligible fashion and written in standard English?

Reviewer #1: No

Reviewer #2: Yes

5. Review Comments to the Author

Reviewer #1: I selected 'partly' for the first question given my question in my review about the search terms used for identifying relevant articles. Depending on the answer to that question, I would change my selection to either yes or no.

I selected 'yes' for the second question, with the small caveat of whether odds ratios are the best way to examine the relationships between continuous variables (see my review).

I selected 'yes' for the third question as the authors indicated that all data is freely available.

I selected 'no' for the fourth question given that "PLOS ONE does not copyedit accepted manuscripts, so the language in submitted articles must be clear, correct, and unambiguous.". While I would describe the manuscript as intelligible, I also think, as I note in my review, that there are many parts that require further clarification.

The manuscript “Adherence to COVID-19 prevention measures and associated factors in Ethiopia: A systematic review and meta-analysis” (PONE-D-21-32366) estimated the rate of adherence to COVID-19 prevention measures in Ethiopia. To do so, the authors conducted a systematic review of relevant literature, concluding that the adherence rate was ~41%, with adherence predicted by sex, age, perceived severity, attitude, and knowledge.

My research experience has focused on health behavior more than systematic reviews, as the focus of my review shows. This paper may further benefit from a reviewer with more experience conducting systematic reviews.

I found the research question important and worthy of investigation. Additionally, I appreciated the authors’ contextualization of the study findings within the broader literature, as well as comparing Ethiopia to other national contexts when discussing the relatively low rate of adherence.

I also have concerns about the paper which, at least until they are addressed, prevent me from recommending publication. These are listed below, though in no particular order.

1. As is, the introduction does not adequately introduce the topic or prepare the reader for the study. It often reads like a list of facts, rather than a coherent perspective on the research topic. Some reorganizing and rewriting should be done to help guide the reader from the general topic to the specific research question. Relatedly, toward the end of the introduction, the authors raise the possibility that “non-adherence to COVID-19 prevention measures could have a negative impact on the health and socio-economy of Ethiopia” (p. 4). While this is an important question, it is not one addressed by the present study. This stage of the introduction should be narrowing in on the research question; I recommend the authors move or remove this sentence.

2. Much of the introduction (p. 3-4) lists the results of studies examining adherence to COVID-19 preventive measures in Ethiopia. These studies indicate that adherence rates are relatively low, with some explanations being “resistance to change, lack of community engagement...and lack of continuous community awareness creation” (p. 4). However, in the next paragraph the authors write that “the evidence bases remained inconsistent and inconclusive”. The extant literature seems consistent, so I’m unsure of where the inconsistency is thought to be.

3. According to the supporting information documents, the authors used the search term “prevention measures” to identify relevant articles. It is unclear whether this strategy would also return articles using the term “preventive measures”, which in my understanding is preferred by many people (I published a paper on this topic and was requested by the editors to use “preventive” rather than “prevention”). If so, please clarify. If not, then there may be relevant articles not included in this systematic review.

In addition to these larger concerns, I have some smaller recommendations for improving the paper.

● It is unclear who “we” refers to on p. 2 where the authors write that, “...we continued to face a shortage of medicines...”. Please clarify.

● On p. 2, why is the number of COVID-19 cases and deaths reported as of July 9, 2021 at 5:29 pm? This seems like a very precise time, though no indication is given as to why it was chosen.

● There are some sentences/phrases which could be improved. Some examples are included below:

○ “After a moment of patience...” (p. 3). This is too vague; I recommend the authors state more specifically what this patience refers to.

○ “...the sex of the study participants found a statistically significant factor...” (p. 10). I believe this is intended to mean “...the sex of the study participants was found to be a statistically significant factor”, though please clarify.

○ “...the study participants are either health professionals or general population who could have an impact to lower the pooled effect size of the adherence level” (p. 12). I believe this is intended to mean that non-health professionals would be expected to have lower adherence rates, though please clarify.

● I was surprised to see that such a small percentage (~1%) of the identified articles were included in the final analysis. This is where a reviewer with more experience in systematic reviews would be helpful.

● Sometimes the authors imply causality when discussing correlations. In one example on p. 10, the authors write that “...age was another factor that had a significant impact on the adherence status...”. Words like impact, influence, etc., suggest a causal relationship, but that’s not quite accurate. I recommend changing these wordings to reflect the correlational nature of the evidence.

● In the analyses of factors associated with adherence to COVID-19 prevention measures (starting on p. 10), I am confused about reporting odds ratios when a simple correlation seems to be more informative. This makes sense for categorical variables (i.e., sex), but less so for continuous variables (i.e., age, knowledge, attitudes, and perceived severity). I recommend that the authors report the correlations, or specify why odds ratios are preferred.

● In the limitations section, the authors state that one limitation is that the review contained no qualitative studies “that would have explored the determinants of adherence to COVID-19 prevention measures” (p. 14). It is unclear why qualitative studies would inform this question, as it seems like a quantitative question.

● In the conclusion section, the authors recommend the government of Ethiopia to “mobilize resources to improve the adherence level of the community to the COVID-19 preventive measures and decrease public fatigue” (p. 14). This is first mention of public fatigue outside of the abstract. If this is to be part of the policy recommendations, it should be discussed in the paper.

● Given that perceived severity was examined, I was surprised to see that there was no mention of the health belief model. I think this is fine if the model did not direct the authors’ thinking, though some reference to it may be warranted.

● I found the comparisons between this systematic review and individual studies in other national contexts (p. 12) somewhat unhelpful given the present purpose was to avoid relying on individual studies. Comparing the present findings to other systematic reviews would be more informative.

● In the interpretation section, the authors claim that “the magnitude of adherence to COVID-19 prevention measures in Ethiopia seems lower than the present findings” (p. 2). However, this study conducts a systematic review of these present findings, so it is unclear how there could be a discrepancy between this study and previous studies. Please clarify.

Reviewer #2: This manuscript describes a systematic review and meta-analysis to estimate the adherence to COVID-19 prevention measures in Ethiopia, and to study the socioeconomic factors associated with adherence. In this study several large electronic databases were used to search articles published between 2019 and 2021. In total, 699 articles were identified in the literature search, and finally seven studies were included in the study after a rigorous screening and selection process. The analyses followed a PRISMA standard for meta-analysis, the methodological quality of articles assessed using the Joanna Briggs Institute (JBI) quality assessment tool, and the inter-study heterogeneity was assessed using I^2 statistics.

However, I have some concerns:

1. It would be helpful for the author to evaluate the potential biases (such as publication bias and reporting bias).

2. The literature search in this study is till July 14, 2021. Given the rapid development of COVID-19 related studies, it would be helpful to have an updated literature search.

Overall, the study method is solid and rigorous. This study identified the key factors associated with adherence to COVID-19 prevention measures, which could provide important insights into improving the adherence level of the community.

6. PLOS authors have the option to publish the peer review history of their article (what does this mean?). If published, this will include your full peer review and any attached files.

Reviewer #1: No

Reviewer #2: No

---

## [Author Response · Author response to Decision Letter 0]

20 Aug 2022

Date 18/08/2022

Subject-response to comments given on our review manuscript.

Dear, Editors and Reviewers 

Thank you very much for your kind consideration of our manuscript titled “Adherence to COVID-19 preventive measures and associated factors in Ethiopia: A systematic review and meta-analysis” and with manuscript ID number of PONE-D-21-32366. On behalf of my all authors, I would like to express my great appreciation to you and reviewers. 

As per the constructive comments and suggestions given from the editorial office and reviewers, we have made detail revision and modification to the manuscript. Overall, this version is more detailed and better in terms of language utilization, coherence, format and subject matter. 

In this version, 

-Literature searching strategies and databases are updated, as suggested

-We have included 8 more articles over the existed 7 articles, a total of 15 articles.

-The manuscript is revised according to the author instructions provided by PLOS ONE. 

-All the figures are corrected using the PACE tool and previously missed table is included in this manuscript. 

-Detailed revision has been made in almost all sections of the manuscript. 

-Linguistic errors are managed by experienced academician and native English, one of the authors (Professor Ruth Paterson)

-Suggested terminological changes are incorporated

In brief, all the amendments are explained in the point-by-point response table given below.

I thank you in advance. 

With kindest regards,

Gdiom Gebreheat 

Question, comment, or suggestion Response

Editor Thank you so much for the constructive comments! All the concerns are addressed accordingly.

1. Please ensure that your manuscript meets PLOS ONE's style requirements, including those for file naming 

We have revised our manuscript accordingly. 

-All figures, table and supportive information are updated

-Headings are updated

2. We suggest you thoroughly copyedit your manuscript for language usage, spelling, and grammar. If you do not know anyone who can help you do this, you may wish to consider employing a professional scientific editing service. 

This revised manuscript is updated accordingly. Moreover, we have made detail and repeated revisions for possible linguistic errors. Initially, the manuscript was checked on free online linguistic error checkers such as Grammarly, Ginger and grammarCheck.net. 

-Then, it was also revised by one of the authors who is native to English (Ruth Paterson) for possible linguistic errors.

3. PLOS requires an ORCID iD for the corresponding author 

I have linked my ORCID iD as per the request. 

4. Please amend either the abstract on the online submission form (via Edit Submission) or the abstract in the manuscript so that they are identical 

We have amended the abstract accordingly.

5. Please upload a new copy of Figures 3-9 as the detail is not clear. Please follow the link for more information:

All figures are revised accordingly 

-All figures are updated using PACE 

6. Please include a copy of Table 1 which you refer to in your text on page 7. -Table 1 is provided within the revised manuscript (P.9-14)

Reviewer #1 Thank you so much for the constructive comments! All the concerns are addressed accordingly.

1. As is, the introduction does not adequately introduce the topic or prepare the reader for the study. It often reads like a list of facts, rather than a coherent perspective on the research topic. Some reorganizing and rewriting should be done to help guide the reader from the general topic to the specific research question. Relatedly, toward the end of the introduction, the authors raise the possibility that “non-adherence to COVID-19 prevention measures could have a negative impact on the health and socio-economy of Ethiopia” (p. 4). While this is an important question, it is not one addressed by the present study. This stage of the introduction should be narrowing in on the research question; I recommend the authors move or remove this sentence. 

We have made amendments to the introduction section (P.2-4). 

In particular;

-Reorganizing and rewriting 

-Adding and removing of statements as necessary 

-Moreover, we have made detail and repeated revisions for possible linguistic errors. Initially, the manuscript was checked on free online linguistic error checkers such as Grammarly, Ginger and grammarCheck.net. Then, it was revised by one of the authors who is native to English (Ruth Paterson). 

-quoted statement is removed, as suggested 

2. Much of the introduction (p. 3-4) lists the results of studies examining adherence to COVID-19 preventive measures in Ethiopia. These studies indicate that adherence rates are relatively low, with some explanations being “resistance to change, lack of community engagement...and lack of continuous community awareness creation” (p. 4). However, in the next paragraph the authors write that “the evidence bases remained inconsistent and inconclusive”. The extant literature seems consistent, so I’m unsure of where the inconsistency is thought to be. 

We have made amendments on the last paragraph of the introduction section (P.3-4). 

-Suggested statements and ideas are corrected to be consistent.

3. According to the supporting information documents, the authors used the search term “prevention measures” to identify relevant articles. It is unclear whether this strategy would also return articles using the term “preventive measures”, which in my understanding is preferred by many people (I published a paper on this topic and was requested by the editors to use “preventive” rather than “prevention”). If so, please clarify. If not, then there may be relevant articles not included in this systematic review. 

In this version of the manuscript, we have made a detail search of literature with additional databases and searching terms. During our search for articles, we have used both terms (prevention, preventive). P.4-5

-As suggested, we have also preferred to use the term “preventive” throughout manuscript.

In addition to these larger concerns, I have some smaller recommendations for improving the paper Thank you so much for the constructive suggestions and comments! All the suggestions are incorporated.

It is unclear who “we” refers to on p. 2 where the authors write that, “...we continued to face a shortage of medicines...”. Please clarify. 

We have removed the statement as part of the revision to the introduction section. P.2-4

On p. 2, why is the number of COVID-19 cases and deaths reported as of July 9, 2021 at 5:29 pm? This seems like a very precise time, though no indication is given as to why it was chosen. 

The mortality and morbidity rates of COVID-19 are being reported at daily base. The first version of our manuscript has been more than a year since last updated (July 9, 2021). So, we had reported the data available by then.

-We have made some amendments and update on this version, though. P.2-4

There are some sentences/phrases which could be improved. Some examples are included below:

○ “After a moment of patience...” (p. 3). This is too vague; I recommend the authors state more specifically what this patience refers to.

○ “...the sex of the study participants found a statistically significant factor...” (p. 10). I believe this is intended to mean “...the sex of the study participants was found to be a statistically significant factor”, though please clarify.

○ “...the study participants are either health professionals or general population who could have an impact to lower the pooled effect size of the adherence level” (p. 12). I believe this is intended to mean that non-health professionals would be expected to have lower adherence rates, though please clarify. 

All the suggested statements and terms are corrected and adopted accordingly.

-P.3, line 58-64

-P.17, line 221-222

-P.18, line 256-258 

I was surprised to see that such a small percentage (~1%) of the identified articles were included in the final analysis. This is where a reviewer with more experience in systematic reviews would be helpful. 

We have updated our article searching strategies considering reviewer #2’s suggestion that there might be new publications since submission of the manuscript to PLOS ONE. Our article searching strategies and selection procedures are reproducible, and clearly explained in the method section. In particular, this can be cross-checked on

-Figure 1 (PRISMA flow chart)

-S1 table

-page 4, 5 and 7 of the manuscript 

Sometimes the authors imply causality when discussing correlations. In one example on p. 10, the authors write that “...age was another factor that had a significant impact on the adherence status...”. Words like impact, influence, etc., suggest a causal relationship, but that’s not quite accurate. I recommend changing these wordings to reflect the correlational nature of the evidence. 

All terms are revised, as suggested.

P.17, line 227 & P.18, line 246

In the analyses of factors associated with adherence to COVID-19 prevention measures (starting on p. 10), I am confused about reporting odds ratios when a simple correlation seems to be more informative. This makes sense for categorical variables (i.e., sex), but less so for continuous variables (i.e., age, knowledge, attitudes, and perceived severity). I recommend that the authors report the correlations or specify why odds ratios are preferred. 

It is well-accepted suggestion, but majority of the authors, of the included articles, transformed the continuous variables into categorical variables during their analysis. That means, these suggested variables were reported as a categorical variables, for instance, age (20-30, 30-40, above 40), knowledge (poor vs good), attitude (favorable vs unfavorable),,,. In our analysis, we found them impossible to manage as continuous variables unless the raw data (dataset) of each article is accessed. Eventually, we decided to manage them as they are reported in each article, as categorical variables. In this case, we have reported the findings also in odds ratio.

In the limitations section, the authors state that one limitation is that the review contained no qualitative studies “that would have explored the determinants of adherence to COVID-19 prevention measures” (p. 14). It is unclear why qualitative studies would inform this question, as it seems like a quantitative question. 

The limitation section is reformed accordingly. We have included more articles (15 articles in systematic review and 11 articles in meta-analysis) in this version. Therefore, absence of qualitative papers could not be a concern in this version of the manuscript. As you said, qualitative articles could not be included in meta-analysis but in the systematic review part. P.20

In the conclusion section, the authors recommend the government of Ethiopia to “mobilize resources to improve the adherence level of the community to the COVID-19 preventive measures and decrease public fatigue” (p. 14). This is first mention of public fatigue outside of the abstract. If this is to be part of the policy recommendations, it should be discussed in the paper. 

we have updated the conclusion section accordingly. Plus, we have added a detail in the discussion section concerning public fatigue as well.

P.20-21 

P.19, line 259-261

Given that perceived severity was examined, I was surprised to see that there was no mention of the health belief model. I think this is fine if the model did not direct the authors’ thinking, though some reference to it may be warranted.

As part of the rework in the discussion section, we have included research articles underpinned by Health belief Model (HBM), to support the concept of perceived disease severity. 

P.20, line 293-298.

I found the comparisons between this systematic review and individual studies in other national contexts (p. 12) somewhat unhelpful given the present purpose was to avoid relying on individual studies. Comparing the present findings to other systematic reviews would be more informative. 

We have updated the discussion section as suggested. We have also removed the comparisons of findings with individual studies. Instead, in this version, we have added more systematic review articles in our comparisons (discussion). 

P.18-20

In the interpretation section, the authors claim that “the magnitude of adherence to COVID-19 prevention measures in Ethiopia seems lower than the present findings” (p. 2). However, this study conducts a systematic review of these present findings, so it is unclear how there could be a discrepancy between this study and previous studies. Please clarify. 

Thank you for your recommendations and we have updated this section accordingly. In particular, we have reworded the mentioned statement as “The magnitude of adherence to COVID-19 preventive measures in Ethiopia appeared to be low”. 

P.21, line 305-306

Reviewer #2 Thank you so much for the constructive comments and suggestions! The concerns are addressed as follows:

1. It would be helpful for the author to evaluate the potential biases (such as publication bias and reporting bias). 

-The Publication bias for the articles included in the meta-analysis of Adherence level is explained in text and figures on page 15, Fig2 and Fig3. We have reported the publication using SE against the effect size i.e Adherence level

-We can also report the publication bias in every associated factor undergone meta-analysis if necessary. But, the articles will be still those of already included in the above publication bias reporting. If we do this, the readability of the paper might be affected because the manuscript will have around 17 figures. Instead, we have submitted the RevMan dataset, from which it is easy to see the publication bias of every associated factor when necessary (i.e for sex, age, attitude, knowledge, attitude). 

2. The literature search in this study is till July 14, 2021. Given the rapid development of COVID-19 related studies, it would be helpful to have an updated literature search.

Overall, the study method is solid and rigorous. This study identified the key factors associated with adherence to COVID-19 prevention measures, which could provide important insights into improving the adherence level of the community. 

We have updated our article searching strategies and databases. Accordingly, we found 8 more articles over the existed 7 articles, a total of 15 articles. In this version of the manuscript, therefore, we have included 15 articles in the systematic review. Of these, 11 articles were included in the meta-analysis.

P.4-5

P.7-8, line 153-164

Figure 1, Table 1, S1-S3

Finally, we greatly appreciate the editor and reviewers for their careful and kindly review again. 

We are also very happy to accept any further comments and suggestions. 

Thank you so much again!

---

## [Decision Letter · Decision Letter 1]

14 Sep 2022

Adherence to COVID-19 preventive measures and associated factors in Ethiopia: A systematic review and meta-analysis

PONE-D-21-32366R1

Dear Dr. Gdiom Gebreheat,

We’re pleased to inform you that your manuscript has been judged scientifically suitable for publication and will be formally accepted for publication once it meets all outstanding technical requirements.

Kind regards,

Carlos Alberto Zúniga-González, Ph.D

Academic Editor

PLOS ONE

Additional Editor Comments (optional):

Dear my sincere congratulations!!!!! I have checked that all reviewers' observations were incorporating on the manuscript.

Reviewers' comments:

Reviewer's Responses to Questions

**Comments to the Author**

1. If the authors have adequately addressed your comments raised in a previous round of review and you feel that this manuscript is now acceptable for publication, you may indicate that here to bypass the “Comments to the Author” section, enter your conflict of interest statement in the “Confidential to Editor” section, and submit your "Accept" recommendation.

Reviewer #2: All comments have been addressed

2. Is the manuscript technically sound, and do the data support the conclusions?

Reviewer #2: Yes

3. Has the statistical analysis been performed appropriately and rigorously? 

Reviewer #2: Yes

4. Have the authors made all data underlying the findings in their manuscript fully available?

Reviewer #2: Yes

5. Is the manuscript presented in an intelligible fashion and written in standard English?

Reviewer #2: Yes

6. Review Comments to the Author

Reviewer #2: (No Response)

7. PLOS authors have the option to publish the peer review history of their article (what does this mean?). If published, this will include your full peer review and any attached files.

Reviewer #2: No

---

## [Editor Report · Acceptance letter]

19 Sep 2022

PONE-D-21-32366R1 

Adherence to COVID-19 preventive measures and associated factors in Ethiopia: A systematic review and meta-analysis 

Dear Dr. Gebreheat:

I'm pleased to inform you that your manuscript has been deemed suitable for publication in PLOS ONE. Congratulations! Your manuscript is now with our production department. 

Kind regards, 

on behalf of

Dr. Prof. Carlos Alberto Zúniga-González 

Academic Editor

PLOS ONE